# Impact of Ocean Acidification on the Gut Histopathology and Intestinal Microflora of *Exopalaemon carinicauda*

**DOI:** 10.3390/ani13203299

**Published:** 2023-10-23

**Authors:** Chao Wang, Wanyu Han, Weitao Cheng, Dexue Liu, Weili Wang, Binlun Yan, Huan Gao, Guangwei Hu

**Affiliations:** 1Jiangsu Key Laboratory of Marine Biotechnology, Jiangsu Ocean University, Lianyungang 222005, China; w2457097460@163.com (C.W.); hwy13604327758@163.com (W.H.); cwt18109635480@163.com (W.C.); xue1553906987@163.com (D.L.); wangweili0922@163.com (W.W.); yanbinlun1962@163.com (B.Y.); huanmr@163.com (H.G.); 2Co-Innovation Center of Jiangsu Marine Bio-Industry Technology, Jiangsu Ocean University, Lianyungang 222005, China

**Keywords:** ocean acidification, *Exopalaemon carinicauda*, intestinal microflora

## Abstract

**Simple Summary:**

Ocean acidification can significantly affect marine organisms, but, so far, no studies have examined its effects on the intestinal flora of crustaceans. Therefore, in this study, the impact of ocean acidification on the gut morphology and intestinal flora of *Exopalaemon carinicauda* was explored. It was found that, after exposure to acidified seawater, the integrity of *E. carinicauda*’s intestinal structure was disrupted and there was a significant change in the flora’s composition. However, the control and acidification groups did not differ significantly in terms of the α diversity. Functional prediction of the acidification groups also suggested that pathways related to metabolism were enriched. The findings indicated that ocean acidification led to an imbalance in the intestinal flora of *E. carinicauda*, which would probably influence the absorption of nutrients and aggravate the susceptibility of shrimp to pathogens.

**Abstract:**

Marine crustaceans are severely threatened by environmental factors such as ocean acidification, but, despite the latter’s negative impact on growth, molting, and immunity, its effects on intestinal microflora remain poorly understood. This work studied the gut morphology and intestinal microflora of *Exopalaemon carinicauda*, grown in seawater of different pH levels: 8.1 (control group), 7.4 (AC74 group), and 7.0 (AC70 group). Ocean acidification was found to cause intestinal damage, while significantly altering the microflora’s composition. However, the α-diversity did not differ significantly between the groups. At the phylum level, the relative abundance of Proteobacteria decreased in the acidification groups, while at the genus level, the relative abundance of *Sphingomonas* decreased. *Babeliales* was a prominent discriminative biomarker in the AC74 group, with Actinobacteriota, Micrococcales, Beijerinckiaceae, *Methylobacterium*, and Flavobacteriales being the main ones in the AC70 group. The function prediction results also indicated an enrichment of pathways related to metabolism for the acidification groups. At the same time, those related to xenobiotics’ biodegradation and metabolism were inhibited in AC74 but enhanced in AC70. This is the first study examining the impact of ocean acidification on the intestinal microflora of crustaceans. The results are expected to provide a better understanding of the interactions between shrimp and their microflora in response to environmental stressors.

## 1. Introduction

Ocean acidification has become a major environmental concern due to the increasing annual discharge of CO_2_ into the atmosphere as a result of human activities. As pointed out in the report of the Intergovernmental Panel on Climate Change (IPCC), a reduction of 0.12 has already been noted in the average pH of oceans since 1880 [1], with projections indicating further declines from the current pH of 8.1 to a pH of 7.8 and 7.4 by 2100 and 2300, respectively [2,3].

Calcifying marine organisms, including mollusks, reef-forming corals, and crustaceans, are particularly affected by ocean acidification. For instance, early studies have reported that changes in the pH value of aquaculture waters had adverse effects on the survival rate, feeding, and growth of *Penaeus indicus* and *Paphia mdabarica* [4,5]. Elevated *P*CO_2_ also has a negative effect on the growth of swimming crabs (*Portunus trituberculatus*) [6]. Similarly, exposure to acidified seawater reduced the survival rate and growth of *Palaemon pacificus* [7], while in the case of *Carcinus maenas* and *Asterias rubens*, impaired food intake and poor growth were noted [8]. In addition, raising crabs (*Petrolisthes cinctipes*, *Hyas Araneus*, and *Cancer magister*) in environments of low pH also induced metabolic stress, alongside poor survival rates, growth rates, and hatching rates, as previously reported [9,10,11].

An increasing number of studies have established that ocean acidification may also affect the intestinal flora of marine organisms. The intestinal microbiota represents a complex ecosystem of microorganisms that live in symbiosis with the host, and as such, they contribute to many of the latter’s physiological processes [12]. For example, a healthy intestinal microbiota generates useful metabolites for the host, while conferring protection against colonization by pathogenic microorganisms. In contrast, dysbiosis, characterized by an imbalance in the microbiota, can yield harmful metabolites that not only alter immunity but also heighten susceptibility to diseases [13,14]. It has been reported that in oysters, stress induced by ocean acidification could reduce the abundance of probiotics in their intestinal tracts; thereby, favoring the proliferation of pathogenic microorganisms, which could eventually disrupt the physiology and health of the oysters [15]. Similarly, following exposure to ocean acidification stress, distinct dysbiosis was observed in the intestinal microflora of sea bream, and this was characterized by the absence of Firmicutes from the bacterial communities, along with an increased relative abundance of Proteobacteria at higher CO_2_ levels [16].

Crustaceans play a significant role in world aquaculture and marine ecosystems but they are highly sensitive to ocean acidification. So far, most studies have examined the impact of ocean acidification on the development, molting, and immunity of crustaceans [17,18,19], but none have explored the effects on their intestinal flora.

*E. carinicauda* is a commercially important shrimp species, as compared with other species; it possesses distinct advantages in terms of fast growth, short breeding cycle, and strong adaptability. Such features also make it a suitable organism for research on crustacean biology [20,21,22,23]. Indeed, as a model species of marine crustaceans, *E. carinicauda* is often used in the study of environmental stress, such as salinity [24] and temperature [25]. The current study used *E. carinicauda*, grown in acidified seawater of different pH levels, to explore changes in their gut morphology and the composition of their intestinal microflora. It is expected that the findings will improve current understanding of the adaptation mechanisms of *E. carinicaudato* in acidic environments.

## 2. Materials and Methods

### 2.1. Experimental Design and Sample Collection

Healthy *E. carinicauda* (6.5 ± 0.2 cm in length and 1.426 ± 0.03 g in weight) were randomly selected from a local aquaculture pond at Jiaxin Aquatic Products Company, Lianyungang city, Jiangsu province, China. These shrimp were cultured in three tanks containing 50 L of aerated and filtered seawater. After being allowed to acclimatize for seven days, they were randomly assigned to the following three groups (with three repetitions in each group; each with 30–40 shrimp): the control group (CONT), in which the shrimp were reared in seawater at a pH of 8.10 ± 0.05; the moderate acidification group (AC74), for which the seawater was at a pH of 7.40 ± 0.05; and the severe acidification group (AC70), with a pH of 7.00 ± 0.05. To create the different conditions, ambient air was continuously bubbled into the aquarium tank for the control group, while for the acidification groups, mixtures of air–CO_2_ were used, with the pH also controlled using an acidometer (LICHEN, Shanghai, China). During the experiment, the pH of each aquarium was monitored and maintained manually. The other water parameters, such as temperature (20 ± 1.1 °C), salinity (25 ± 0.5), and dissolved oxygen (5.7 ± 0.2), were maintained and evaluated based on standard methods (APHA, 2005) [26]. Commercial feed was given to the shrimp twice a day (8:00, 18:00). The unfed feed, waste, and dead shrimp were collected daily. And after 30 days of exposure to CO_2_, the survival rate was calculated and a survival curve was drawn. Intestinal tissues, including the contents, were collected from each sampled shrimp under aseptic conditions using sterile tweezers and scalpels. In each case, three intestines were put together as one sample. The 17 samples (CONT: 6 samples, AC74: 6 samples, AC70: 5 samples) were then rapidly frozen in liquid nitrogen prior to storage at −80 °C, until required for DNA extraction.

### 2.2. Assessment of Intestinal Morphology

After 30 days of exposure to CO_2_, the intestines of three shrimps from each tank were collected. A 24-h tissue fixation process was then performed at 4 °C using 4% paraformaldehyde, and this was followed by sample dehydration using varying ethanol concentrations (75% alcohol for 4 h, 85% alcohol for 2 h, 90% alcohol for 2 h, 95% alcohol for 1 h, anhydrous ethanol I for 30 min and anhydrous ethanol II for 30 min). The samples were subsequently subjected to acetone and xylene treatment to make the tissues transparent, and after being embedded in paraffin, 3-μm thick sections were cut. The latter were eventually stained with hematoxylin and eosin (HE) before being observed under a microscope (OLYMPUS, DP26).

### 2.3. Extraction of DNA and Sequencing

Genomic DNA was extracted prior to high-throughput sequencing, as previously described [27]. Briefly, this involved DNA extraction using the Magnetic Soil and Stool DNA Kit (TIANGEN^®^, Beijing, China), according to the manufacturer’s instructions, followed by assessment of the DNA quality using agarose gel electrophoresis. The V3-V4 region of the 16S rRNA gene was amplified using specific primers (341F 5′ CCTAYGGGRBGCASCAG 3′ and 806 R 5′ GGACTACNNGGGTATCTAAT3′) [28], then sequenced on the NovaSeq 6000 Illumina platform (San Diego, CA, USA). In order to obtain high-quality clean reads, raw reads were filtered according to the following rules: (1) we used CUTADAPT 3.3 software to intercept and filter data, and the minimum sequence length was 20 bp; (2) FLASH [29] (Mago and Salzberg 2011) (1.2.11) software was used to splice the double-ended data, with the length of min-overlap being 10 bp and the max mismatch density being 0.2; (3) FLASH (1.2.11) software was used to control the quality of the spliced data, the qualified quality Phred was 19 and the unqualified percent limit was 15.

### 2.4. Analysis of Microbial Diversity and Predicted Functions

Analysis of amplicon sequencing data was performed with QIIME 2 (version QIIME2-202006, (https://www.qiime2.org, accessed on 15 March 2022), with the DADA2 plugin first used for denoising single-end demultiplexed reads. The resulting operational taxonomic units (OTUs) [30] were subsequently classified into taxa with a naïve Bayes classifier trained on the SILVA v132 database (clustered at 99% identity, with only the V3–V4 region of 16S rRNA sequences extracted) (https://www.arb-silva.de, accessed on 15 March 2022). Shared and unique OTUs were eventually identified using Venn diagrams, while relative abundances of the different taxa were displayed as stacked bar plots that were generated after using the “sum” mode to group the relative abundances of replicate samples.

Phylogenetic diversity was analyzed with FastTree [31], and, prior to additional diversity analysis, the dataset was standardized via rarefying the feature table to the lowest number of reads in the samples. The q2-diversity plugin was then used to calculate α- and β-diversity metrics, namely the Shannon diversity index, Chao1 index, Good’s coverage, and Bray–Curtis distances. The dissimilarity between microflora structures was also determined using principal coordinate analysis (PCoA) and analysis of similarity (ANOSIM) based on the Bray–Curtis distances. In addition, differentially abundant taxa (biomarkers) between the groups were identified using linear discriminant analysis (LDA) effect size (LEfSe) with an LDA threshold of >3.5 [32]. The functional profiles of the intestinal microflora were eventually predicted using PICRUSt2 (https://github.com/picrust/picrust2/, accessed on 15 March 2022), with the STAMP package (version 2.1.3) subsequently used to statistically analyze and visualize the results.

### 2.5. Statistical Analysis

The value of each variable was presented as the mean ± SE. SPSS (Ver 17.0) was then used for one-way analysis of variance (ANOVA), as well as Duncan multiple range tests. All data were firstly examined for homogeneity of variance using SPSS version 17.0. Results with *p*-values < 0.05 were considered to be statistically significant.

## 3. Results

### 3.1. Survival Rate, Histological Structure of Intestinal Tissues

In the present study, the survival of *E. carinicauda* was significantly reduced in pH 7.4 and pH 7.0 compared to control pH 8.1 (Appendix A). This work studied the impact of acidification stress on the integrity of *E. carinicauda*’s intestinal tissues. For the control group, the intestinal mucosa had a normal morphology (Figure 1A), while for the AC74 and AC70 groups, exposure to the lower pH induced obvious pathological alterations. In particular, deformation, the presence of cavitation bubbles, and exfoliation from the basement membrane were observed for the epithelial cells (Figure 1B,C).

### 3.2. Characteristics of the High-Throughput Sequencing Data

From 17 samples, a total of 2,029,062 raw reads were obtained, with subsequent merging and quality control yielding a total of 1,787,879 raw tags (Appendix A). Potential chimeric sequences were then removed with vsearch (2.16.0), resulting in an average number of 94,456 clean tags per sample (a total of 1,605,753 clean sequences). Following data filtering, 4657 OTUs were identified, of which 769 were common to the three groups. The results also showed that the control, AC74, and AC70 groups had 941, 1070, and 987 unique OTUs, respectively (Figure 2A). In addition, rarefaction curves based on OTU numbers were close to the horizontal state, suggesting that the sequencing data for each sample sufficiently reflected intact bacterial communities (Figure 2B). For all samples, the Good’s coverage index was 0.9989, thereby suggesting an adequate sequencing depth.

### 3.3. Diversity of the Intestinal Microflora

α-diversity metrics, reflecting the abundance and diversity of microbial populations, were calculated for each group to determine how *E. carinicauda*’s intestinal microflora was affected by OA stress. In this case, the control, AC74, and AC70 groups had a Chao1 index of 631.691, 717.876, and 758.405, respectively, while their Shannon index was 5.262, 6.293, and 6.584, respectively (Table 1). However, even though the two indices varied between the three groups, they were not significantly different. Furthermore, the PCoA plot indicated that exposure to OA stress induced significant changes to the intestinal microflora’s structure (Figure 3).

### 3.4. Community Structures of the Intestinal Microflora

For all groups (i.e., control and acidification), the intestinal microflora consisted of Proteobacteria, Firmicutes, Actinobacteriota, and Bacteroidota as the dominant phyla (Figure 4). In comparison with the control, AC74 and AC70 had a significantly higher relative abundance of Actinobacteriota (Figure 5A). On the other hand, there was a decrease in the abundance of Proteobacteria in an OA stress-dependent manner (Figure 5B).

At the genus level, *Sphingomonas*, *Photobacterium*, and *Lactococcus* were dominant in the three groups (Figure 6). More specifically, the AC74 and AC70 groups were significantly enriched in *Photobacterium* compared with the control. In contrast, OA stress significantly decreased the relative abundance of *Sphingomonas* from 15.06% for the control to 0.13% and 1.77% for AC74 and AC70, respectively. In a similar way, the lower pH decreased the relative abundance of *Lactococcus*, with relative abundances of 14.89%, 8.41%, and 6.90% for the control, AC74, and AC70 groups, respectively.

LEfSe analysis using an LDA threshold of >3.5 identified 34 differentially abundant (biomarker) taxa between the three groups (Figure 7). Six of these were enriched in the control, with Sphingomonadaceae and Chitinophagales being the most significant ones. Similarly, for the AC74 group, 22 biomarker taxa were found, with the top 5 including Babeliales, Dependentiae, Babeliae, *Vermiphilaceae*, and *Sulfitobacter*. Finally, the AC70 group had six differentially abundant taxa, namely Actinobacteriota, Micrococcales, Beijerinckiaceae, *Methylobacterium*, Flavobacteriales, and Myxococcota.

### 3.5. Functional Prediction of the Intestinal Microflora

PICRUSt2 analysis indicated that 15 MetaCyc pathways were significantly different between the control and AC74 groups (*p <* 0.05) (Figure 8A). In particular, the pathways associated with environmental information processing, genetic information processing, metabolism, amino acid metabolism, energy metabolism, and lysine biosynthesis were enriched in the AC74 group compared with the control, while those associated with metabolism, arginine and proline metabolism, carbohydrate metabolism, pentose and glucuronate interconversions were blocked. There were also two pathways that differed significantly between the control and AC70 groups (*p <* 0.05) (Figure 8B), with the latter enriched in metabolism, naphthalene degradation, metabolism of terpenoids, and polyketides, as well as pathways related to tetracycline biosynthesis.

## 4. Discussion

The study of gut microbiota has attracted significant interest in recent years, as it influences the ability of marine organisms to adapt to environmental conditions. Crustaceans, being calcifying organisms, are especially susceptible to the effects of ocean acidification. However, it is still unclear how their gut microbiota responds to stress induced by ocean acidification. Therefore, using a high-throughput sequencing approach, this work seeks to determine how *E. carinicauda*’s intestinal microflora is affected by ocean acidification stress.

The ability of shrimp to absorb nutrients is largely dependent on their intestinal epithelium and microvilli. Thus, the structural integrity of the intestinal mucosa is crucial for their health [33]. However, it has been reported that environmental stress could damage that mucosal layer in the intestine [33,34], with previous studies showing obvious damage to the intestinal mucosal epithelial cells of *Litopenaeus vannamei* after high and low pH stress [35]. Similarly, the current work showed clear pathological damage to the intestinal mucosa of the AC74 and AC70s groups after exposure to acidified seawater, hence indicating that ocean acidification could disrupt both the structural integrity and the digestive functions of the intestine.

For aquatic animals, environmental stressors can alter the diversity of their intestinal microbial flora, a feature that is closely related to their health [36,37]. For instance, it has been reported that an increase in water temperature can significantly decrease the α-diversity of intestinal microbial communities in mussels [38]. Similarly, in the case of *Macrobrachium nipponense*, the α-diversity decreased after exposure to hypoxic stress for four weeks [39]. However, in this study, the groups (i.e., control and acidification) did not differ significantly in terms of their α-diversity metrics, with similar results reported for the Pacific oyster *Crassostrea gigas* [15]. On the other hand, β-diversity analysis confirmed obvious differences in the structure of the intestinal microflora for *E. carinicauda* reared at pH 7.4 and 7.0. Increasing evidence shows that crustaceans could be more susceptible to pathogenic infections when their intestinal flora is disrupted. Therefore, the dysbiosis observed in *E. carinicauda*’s intestinal microflora after exposure to ocean acidification may facilitate the proliferation of opportunistic pathogens and compromise their health.

This work also found that the control and acidification groups differed significantly in terms of the intestinal microflora composition. Specifically, the relative abundance of Proteobacteria and the Firmicutes/Bacteroides ratio decreased with increasing acidification stress. Proteobacteria typically dominate the intestinal flora of marine shrimp [40,41], and studies have shown that a decrease in their relative abundance is often correlated with an imbalanced or unstable intestinal flora. Therefore, Proteobacteria are often less abundant in the intestine of shrimp infected by pathogens or experiencing slow growth [42,43]. Such decrease in the abundance of Proteobacteria may also hinder the absorption of nutrients in *E. carinicauda*, thus hampering the latter’s ability to uptake nutrients under acidification stress. The Firmicutes/Bacteroides ratio is considered to be significantly correlated to the composition of the intestinal flora. For instance, it has been reported that the relative abundance of *Bacteroidetes* and Firmicutes in shrimp intestine is not related to carbohydrate degradation efficiency. However, a higher ratio of Firmicutes to Bacteroides translates into enhanced intestinal transport and ability to digest nutrients [44], with this condition being easily influenced by the culture environment and the nutritional composition of the feed [45]. In this study, the acidification groups had a lower Firmicutes/Bacteroides ratio, which suggests a reduced ability for intestinal transport and digestion in *E. carinicauda* under acidification stress.

At the genus level, *E. carinicauda*’s intestine was dominated by *Photobacterium* and *Sphingomonas*, as previously reported in the case of *L. vannamei* [46] and *Panulirus Homarus* [47]. *Photobacterium* is a marine bacterium of the *Vibrio* family, and while some members of this genus are known to be pathogenic [48], other species have more symbiotic relationships with their hosts, which include fish and crustaceans [49]. It was previously reported that the abundance of luminescent bacteria in the intestines of Nile tilapia and *L. vannamei* increased after salinity stress [50]. Similarly, the current work observed an increase in the abundance of *Photobacterium* after exposure to acidified seawater, which indicates that *Photobacterium* was a symbiotic bacterium but pathogenic under the experimental conditions. Liu et al. (2016) [51] investigated two pathogenic *Photobacterium* strains isolated from *E. carinicauda*, with both being responsible for mortality in *E. carnicauda* and *L. vannamei*. Their results also suggested that *Photobacterium* was a common shrimp pathogen, in a similar way to *Vibrio parahaemolyticus* and *V. alginolyticus*. The current results showed an increase in the abundance of *Photobacterium* with increasing ocean acidification stress, which indicates that such stress could heighten the likelihood of shrimp infection through opportunistic pathogens. *Sphingomonas*, a non-spore-forming Gram-negative bacterium, is ubiquitous in nature and is considered to be a conditional pathogen that may cause human infections [52]. Therefore, the observed abundance of these genera indicates that ocean acidification may increase the chances of opportunistic bacteria or pathogenic bacteria adhering to or penetrating the damaged intestinal epithelium of shrimp, thus damaging the latter’s intestinal immunity. These results also showed that the acidification of oceans can decrease the relative abundance of probiotics, hence affecting the growth of shrimp while promoting the proliferation of pathogenic species in shrimp intestines.

Changes in *E. carinicauda*’s intestinal microflora composition in response to ocean acidification stress were also associated with functional changes. The predicted functions revealed that, for the AC74 group, metabolism, amino acid metabolism, and energy metabolism were the main pathways to be significantly enriched, which indicated that ocean acidification stress induced a rapid proliferation of the intestinal microflora. An enrichment of similar pathways involved in the proliferation of microflora, namely metabolism, naphthalene degradation, metabolism of terpenoids and polyketides, and tetracycline biosynthesis pathway, were also noted for the AC70 group. The increased consumption of nutrients as a result of microbial proliferation may eventually lead to host malnutrition [15]. According to previous reports, environmental mutations and exposure to environmental pollutants may also significantly alter the intestinal microbiota of aquatic animals [34,50,53,54,55]; thus, inducing inflammation and immune disorders in the host [56].

It should be noted that, in the AC74 group, pathways related to arginine and proline metabolism, carbohydrate Metabolism, and glycolysis were significantly inhibited. In particular, the lower catabolism of carbohydrates via the intestinal microflora could provide the host with an additional energy source for coping with the ocean acidification stress [27]. However, enhanced naphthalene degradation was also noted for the AC70 group, which indicated that, under ocean acidification stress, the microflora would consume the energy source within *E. carinicauda*’s intestine more rapidly. This, in turn, would negatively impact the health of *E. carinicauda*.

## 5. Conclusions

A decrease in seawater pH damaged the morphological structure of *E. carinicauda*’s intestine, resulting in a digestive disorder. The lack of important functional phyla, such as Proteobacteria, may contribute to an imbalance in the intestinal flora, thereby adversely affecting the absorption of nutrients. Additionally, the excessive growth of potential pathogens and the resulting nutrient consumption may cause *E. carinicauda* to be more susceptible to diseases. Finally, with more severe ocean acidification stress, the energy demand of the intestinal flora may increase, leading to a potential energy deficit in *E. carinicauda*, thus posing significant challenges to its overall health.

## Figures and Tables

**Figure 1 animals-13-03299-f001:**
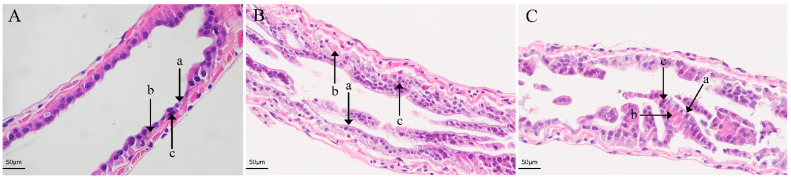
Intestinal tissues of *E. carinicauda* that had been stained with HE after exposure to different pH conditions for 30 days: (**A**) the control group (CONT), ×400; (**B**) the pH 7.4 group (AC74), ×400; (**C**) the pH 7.0 group (AC70), ×400; (a) indicates the brush border; (b) shows the epithelium; and (c) represents the nuclei.

**Figure 2 animals-13-03299-f002:**
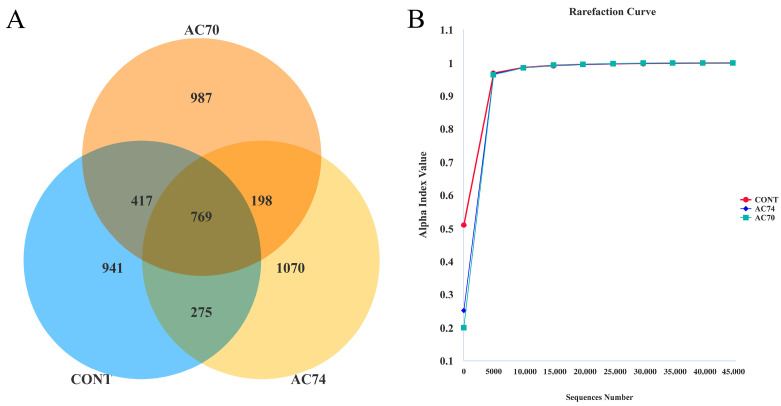
(**A**) Venn diagram showing the three groups (CONT, AC74, and AC70), along with the number of shared and unique OTUs. (**B**) Rarefaction curves: the horizontal axis represents the amount of sequencing data, and the vertical axis represents the corresponding alpha diversity index.

**Figure 3 animals-13-03299-f003:**
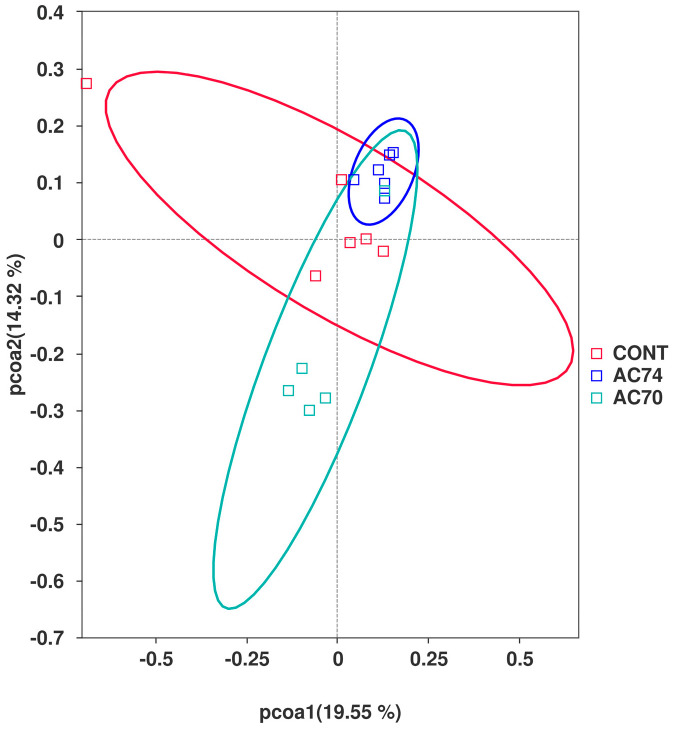
PCoA analysis of the intestinal microflora for the three groups (CONT, AC74, and AC70) based on Bray–Curtis distances.

**Figure 4 animals-13-03299-f004:**
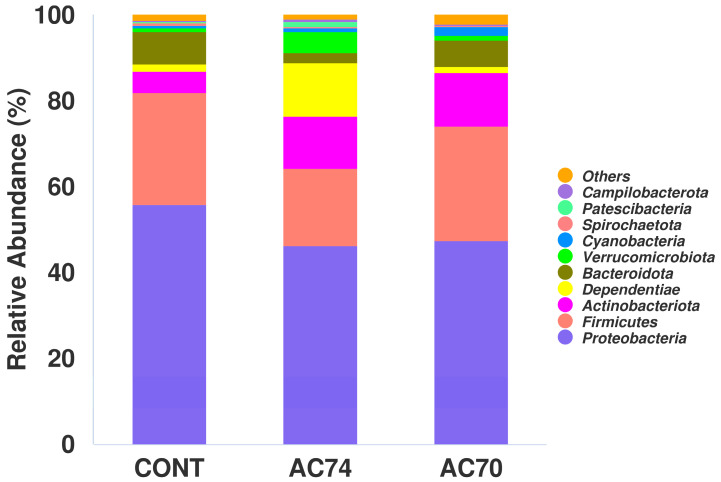
Relative abundance of different phyla in the intestinal microflora of the three groups (CONT, AC74, and AC70). The figure shows only the top ten abundant phyla, with the rest grouped as “Others”.

**Figure 5 animals-13-03299-f005:**
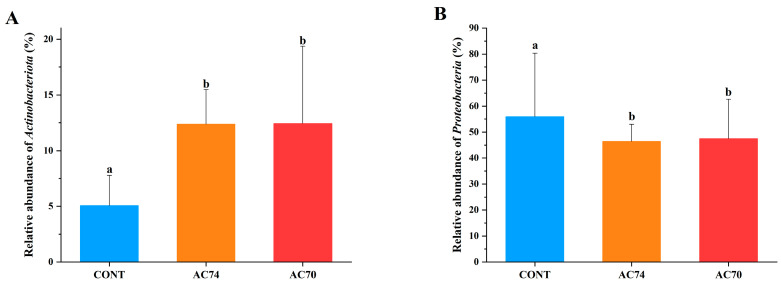
Relative abundance of major phyla in each group: (**A**) relative abundance of Actinobacteriota in each group; (**B**) relative abundance of Proteobacteria in each group. Significant differences (*p* < 0.05) between the groups are indicated by the lowercase letters (a, b).

**Figure 6 animals-13-03299-f006:**
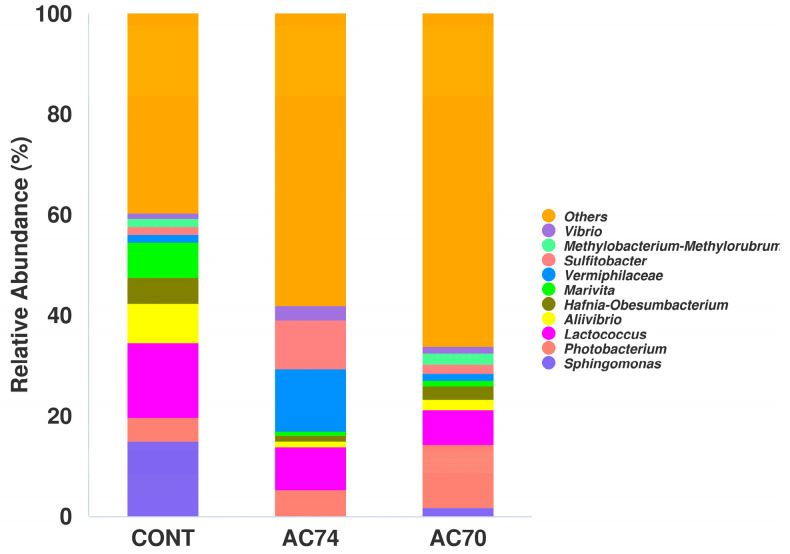
Relative abundance of different genera in the intestinal microflora of the three groups (CONT, AC74, and AC70). The figure shows only the top ten abundant genera, with the rest grouped as “Others”.

**Figure 7 animals-13-03299-f007:**
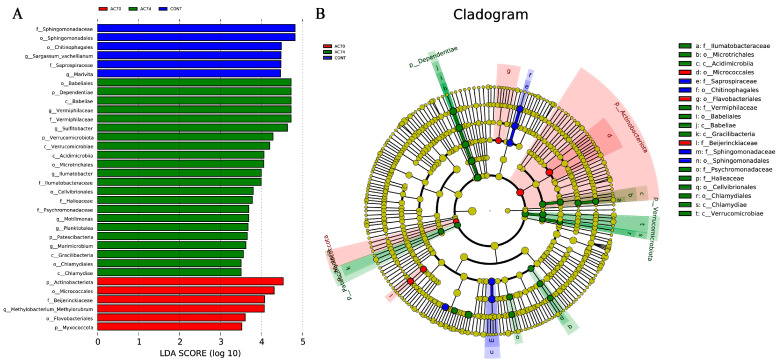
Results of LEfSe analysis showing the differentially abundant taxa between the three groups (CONT, AC74, and AC70) using an LDA threshold of >3.5: (**A**) histogram of LDA distribution; (**B**) cladogram based on the LEfSe analysis.

**Figure 8 animals-13-03299-f008:**
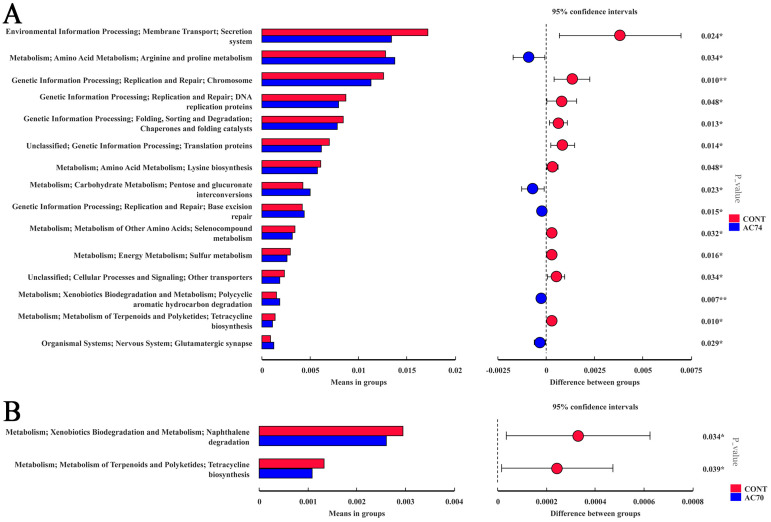
The functional pathways showing significant differences between the control and AC74 groups (**A**), and the control and AC70 groups (**B**), based on PICRUSt2 and STAMP analysis. The *p*-values are based on Welsh’s *t*-test and corrected with Benjamini–Hochberg FDR. The symbol * is *p*-value > 0.01, and ** is *p*-value ≤ 0.01.

**Table 1 animals-13-03299-t001:** α-diversity metrics of the intestinal microflora for the three groups (CONT, AC74, and AC70).

Alpha Diversity	Mean			Standard Deviation			Significance
	CONT	AC74	AC70	CONT	AC74	AC70	
Chao 1	631.691	717.876	758.405	168.187	170.615	375.091	*p* = 0.686
Shannon	5.262	6.293	6.584	0.768	0.584	1.491	*p* = 0.125

## Data Availability

Not applicable.

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
