# Peer review of "Impact of Ocean Acidification on the Gut Histopathology and Intestinal Microflora of Exopalaemon carinicauda"

_animals, 2023, doi:10.3390/ani13203299_

Round 1

Reviewer 1 Report

I suggest that the title be adjusted as it creates high expectations about the scope of their research and the results you are showing.

Effects of decreasing water pH on the gut histopathology and intestinal microflora of Exopalaemon carinicauda.

Simple summary

This section should be adjusted following my comments below.

Abstract

This section should be adjusted following my comments below.

Introduction

This section is well prepared.

Line 46 “et al,” eliminate.

Material and methods

I have a couple of observations regarding your research.

1) Although the process of pH drop is called acidification, the treatments you are applying are Control (8.1), AC74 (7.4), and AC70 (7.0), so you are looking at the effect of two alkaline and one neutral pH treatments. It would be worthwhile to clarify this aspect in your paper.

2) You indicate that to maintain the values of 7.4 and 7.0 pH, you injected a mixture of air and CO2; however, injecting CO2 not only acidifies the medium but can lead to hypoxic conditions, significantly affecting the microbiome. How did you ensure that the decrease in dissolved oxygen did not affect the microbiome of your shrimp, rather than lowering the pH to alkaline and neutral conditions?

Did you use replicates? How many shrimps per replicate did you use? What is the size (mean weight and total length) of the shrimps? Did you use an open or close system? Which other water quality parameters did you measure? What happens with the alkalinity, dissolved oxygen, salinity, ammonia, nitrate, nitrite, etc.? Include more detail in shrimp handling.

How many shrimps per replicate did you use for the metagenome analysis?

Statistical analysis

Did you test normality and homoscedasticity of your data? They are important postulates for the parametric statistics and avoid Errors type I and II. Clarify this point.

Results

The histological analysis could be enriched by measuring anatomical structures and their respective statistical analysis. I request the authors to include an additional table.

The authors must include zootechnical information about their experiment, such as growth, survival, feed conversion factor, specific growth rate, etc. Include a table and make the corresponding statistics. Make the proper adjustments in Material and methods section.

This research would be greatly enriched if the authors included blood biochemical parameters, immune gene expression or the measurement of antioxidant enzymes, which may be important indicators of the effect of pH depletion.

Discussion

I suggest that authors adjust this section according to the scope of their research. As I commented previously, ocean acidification is a reality; however, their study focuses on the decrease in pH from 8.1 to 7.4 and 7.0, which causes them to have one treatment under alkaline and one neutral condition. This aspect is crucial, although the evidence on the long-term effects cannot be visualized in a 30-day experiment, in addition to the fact that shrimp, being mobile organisms, can migrate to more stable zones from the ecological point of view and the aquaculture point of view, all water quality parameters must be taken care of. I am concerned that you do not refer to the water quality of your experiment, particularly the O2 levels, since they could be stressing the animals in hypoxic conditions.

Lines 246-247 “is affected by ocean acidification stress.” Change this sentence to "the decrease of pH in alkaline and neutral conditions".

It is unclear to me the damage to the intestinal epithelium, so you need to include measurements and discuss them to know if there is really an effect of the pH decrease in your experiment.

You need to include growth and survival parameters from your study to give more context to practical variables used in aquaculture.

Avoid the excessive use of “ocean acidification” in your MS.

Conclusion

Adjust this section according to the above comments.

Author Response

Dear editor and reviewers

Thank you for dedicating your time to reviewing our manuscript (animals-2642524). We sincerely appreciate the valuable comments and suggestions provided by the reviewers, which have greatly contributed to the improvement of our work. We have diligently incorporated your kind and constructive feedback into the revised version of the manuscript, as evidenced by the itemized responses provided below and the corresponding revisions/corrections in the re-submitted files.

We are grateful for the opportunity to submit a revised copy of the manuscript and remain hopeful that it will be accepted for publication in animals.

Sincerely yours,

Guangwei Hu

On behalf of all co-authors

# Reviewer 1

Comment 1: I suggest that the title be adjusted as it creates high expectations about the scope of their research and the results you are showing. Effects of decreasing water pH on the gut histopathology and intestinal microflora of Exopalaemon carinicauda.

Response 1: Thanks for your suggestions. Ocean acidification is the decrease in the pH of the Earth's ocean. Between 1950 and 2020, the average pH of the ocean surface fell from approximately 8.15 to 8.05, with projections indicating further declines from the current pH of 8.1 to a pH of 7.8 and 7.4 by 2100 and 2300, respectively. Decreased ocean pH has a range of potentially harmful effects for marine organisms. These include reduced calcification, depressed metabolic rates, lowered immune responses, and reduced energy for basic functions such as reproduction. Therefore, we would prefer to present ocean acidification prominently. Thanks again for the above comment.

Comment 2:

Simple summary

This section should be adjusted following my comments below.

Response 2: Thanks for your comment please see our revised manuscript and response 6

Comment 3:

Abstract

This section should be adjusted following my comments below.

Response 3: Thanks for your comment, please see our revised manuscript and response 6

Comment 4:

Introduction

This section is well prepared.

Response4: Thanks very much for your positive comment.

Comment 5: Line 46 “et al,” eliminate.

Response 5: Agree. Line 46 “et al, "has been deleted. Please see line 46.

Comment 6: I have a couple of observations regarding your research. 1) Although the process of pH drop is called acidification, the treatments you are applying are Control (8.1), AC74 (7.4), and AC70 (7.0), so you are looking at the effect of two alkaline and one neutral pH treatments. It would be worthwhile to clarify this aspect in your paper.

Response 6: Thank you for your comment. It appears that there may have some misconceptions regarding ocean acidification and our explanation in this section. Firstly, ocean acidification refers to a decline in the pH of the ocean primarily due to the absorption of carbon dioxide (CO2) from the atmosphere, and it should not be confused with the concept of "decrease in water pH". Secondly, the average pH of the ocean's surface water is 8.1. This is why we designated pH 8.1 as the control group, and the other two as the acidification groups. Therefore, our focus is not on the impact of two alkaline and one neutral pH treatments, but rather on the effects of different levels of acidification.

Comment 7: You indicate that to maintain the values of 7.4 and 7.0 pH, you injected a mixture of air and CO2; however, injecting CO2 not only acidifies the medium but can lead to hypoxic conditions, significantly affecting the microbiome. How did you ensure that the decrease in dissolved oxygen did not affect the microbiome of your shrimp, rather than lowering the pH to alkaline and neutral conditions?

Response 7: We appreciate your comment, as your concern holds significant importance for our experiment. In order to ensure consistency, we diligently regulated the dissolved oxygen levels of both the control group (pH 8.1) and the experimental groups (pH 7.4 and 7.0) at a constant range of 5-6 mg/L. This was achieved by adjusting the oxygen valve, as detailed in the Materials and Methods section of the revised manuscript. Please see line 97-100.

Comment 8: Did you use replicates? How many shrimps per replicate did you use? What is the size (mean weight and total length) of the shrimps? Did you use an open or close system? Which other water quality parameters did you measure? What happens with the alkalinity, dissolved oxygen, salinity, ammonia, nitrate, nitrite, etc.? Include more detail in shrimp handling.

Response 8: Thank you for your feedback. The experiment was conducted with three groups (pH8.1, pH7.4, and pH7.0), each consisting of three replicates. Each replicate contained 30-40 shrimps. The average length of the shrimp was measured to be 6.5±0.2cm, while the weight was recorded as 1.426±0.03g. Open system (tanks) were utilized for the experiment, and various water quality parameters including salinity, temperature, and dissolved oxygen were measured. Further details regarding this aspect can be found in the materials and methods section. Please refer to lines 86-91 and 97-100.

Comment 9: How many shrimps per replicate did you use for the metagenome analysis?

Response 9: Thank you for your comment. For 16S sequencing and analysis, totally 17 samples were used, three shrimps in each sample (replicate). 6 samples for CON8.1 group, 6 samples for AC7.4 group. Due to the limited number of shrimps in AC70 group (pH7.0), 5 samples were used. We have clarified this in the revised manuscript, please see line 103-106.

Comment 10: Did you test normality and homoscedasticity of your data? They are important postulates for the parametric statistics and avoid Errors type I and II. Clarify this point.

Response 10: Before ANOVA, normality and homoscedasticity were analyzed. We have amended this part in the revised manuscript. Please see the revised version, line 157.

Comment 11: The histological analysis could be enriched by measuring anatomical structures and their respective statistical analysis. I request the authors to include an additional table.

Response 11: Thank you for your suggestion. The objective of this study is to investigate the integrity of E. carinicauda's intestinal tissues under various treatments. The findings depicted in figure 1 indicate that the control group exhibited a normal and intact morphology of the intestinal tissue, whereas the AC74 and AC70 groups displayed deformation and exfoliation of epithelial cells from the basement membrane. It is worth noting that previous research (cited below) has commonly presented histological results without accompanying statistical analysis, which is considered acceptable in the field.

References for review

[1] Cheng, C., Ma, H., Liu, G., et al. Toxic effects of cadmium exposure on intestinal histology, oxidative stress, microbial community, and transcriptome change in the mud crab (Scylla paramamosain). Chemosphere. 2023, 326, 138464

[2] Duan, Y., Wang, Y., Liu, Q., et al. Changes in the intestine barrier function of Litopenaeus vannamei in response to pH stress. Fish & Shellfish Immunology. 2019, 88, 142-149.

Comment 12: The authors must include zootechnical information about their experiment, such as growth, survival, feed conversion factor, specific growth rate, etc. Include a table and make the corresponding statistics. Make the proper adjustments in Material and methods section.

Response 12: We express our gratitude to the reviewer for bringing this matter to our attention. It is true that we should have included the zootechnical information in our study. Nonetheless, it is important to note that the subjects of our investigation were adult shrimps, whose growth is not expected to undergo substantial changes within a 30-day period, as supported by practical experience and previous research. This holds true for both the feed conversion factor and the specific growth rate. In light of this, we have incorporated additional survival data in the revised manuscript, as it would provide more informative insights for this particular experiment. Please see line 161-162 and Figure F1 supplementary materials.

Comment 13: This research would be greatly enriched if the authors included blood biochemical parameters, immune gene expression or the measurement of antioxidant enzymes, which may be important indicators of the effect of pH depletion.

Response 13: We appreciate your comment and concur with your suggestion that the research would benefit from the inclusion of blood biochemical parameters, immune gene expression, and antioxidant enzymes. However, it is important to note that these aspects have already been extensively studied in previous research (Comparative Biochemistry and Physiology, Part C 239 (2021) 108843). Our primary objective is to elucidate the effects of ocean acidification on the gut histopathology and intestinal microflora in Exopalaemon carinicauda. Consequently, we did not incorporated these particular elements into our study. It is worth mentioning that the intestinal microbiota plays a significant role in various physiological processes of the host, we are currently investigating the immune genes associated with the intestinal microflora, and the relevant results will be published in the future.

Comment 14: I suggest that authors adjust this section according to the scope of their research. As I commented previously, ocean acidification is a reality; however, their study focuses on the decrease in pH from 8.1 to 7.4 and 7.0, which causes them to have one treatment under alkaline and one neutral condition. This aspect is crucial, although the evidence on the long-term effects cannot be visualized in a 30-day experiment, in addition to the fact that shrimp, being mobile organisms, can migrate to more stable zones from the ecological point of view and the aquaculture point of view, all water quality parameters must be taken care of. I am concerned that you do not refer to the water quality of your experiment, particularly the O2 levels, since they could be stressing the animals in hypoxic conditions.

Response 14: Thank you for your advice. During the experiment, we measured the water quality factors such as salinity, temperature and dissolved oxygen. This part has been added in materials and methods. Please refer to lines 97-100.

Comment 15: Lines 246-247 “is affected by ocean acidification stress.” Change this sentence to "the decrease of pH in alkaline and neutral conditions".

Response 15: Thanks for your comment. we would prefer to present ocean acidification prominently. Please see the explanation in response 6.

Comment 16: It is unclear to me the damage to the intestinal epithelium, so you need to include measurements and discuss them to know if there is really an effect of the pH decrease in your experiment.

Response 16: We express our gratitude for your comment. In this study, we conducted an investigation into the impact of acidified seawater on the structural integrity of intestinal tissue. In comparison to the control group, which exhibited unaltered and undamaged intestinal tissue, we observed deformations in the intestinal tissue and the detachment of epithelial cells from the basement membrane (Figure 1). We have thoroughly discussed these findings in the manuscript. Please see line162-168.

Comment 17: You need to include growth and survival parameters from your study to give more context to practical variables used in aquaculture.

Response 17: We supplemented the survival data in the revised manuscript. Please refer to Figure F1 in the supplementary materials and response 12.

Comment 18: Avoid the excessive use of “ocean acidification” in your MS.

Response 18: Ocean acidification refers to a decline in the pH of the ocean primarily due to the absorption of carbon dioxide (CO2) from the atmosphere, and it should not be confused with the concept of "decrease in water pH". Therefore, we would prefer to present ocean acidification prominently in this study. Thanks again for your comment.

Comment 19:

Conclusion

Adjust this section according to the above comments

Response 19: As described in previous responses, we would prefer to present ocean acidification prominently in this study.

Reviewer 2 Report

 Dear Authors 

Please find below my comments.

It is recommended to include the following reference in the introduction.           

Lin, W., Ren, Z., Mu, C., Ye, Y., & Wang, C. (2020). Effects of elevated p CO2 on the survival and growth of Portunus trituberculatus. Frontiers in Physiology, 11, 750.

Line 84: It is suggested to include abiotic data such as temperature, oxygen (saturation and concentration), and water turnover.

Line 93: It is recommended to specify whether tissue and intestinal contents are included.

Line 108: It is suggested to include details of the quality controls performed.

Line 111: It is recommended to provide the reference for the primers used.

Line 140: It is suggested to include a table with the frequency of described findings. Please review possible artifacts due to sample processing or sectioning.

It is possible to include quantitative comparisons of 16S amplicon abundance by comparing the read counts of the analyzed samples in a supplementary figure.

It is possible to include rarefaction curves of 16S amplicons for the different groups in a supplementary figure.

The labels on figures 4, 5, and 6 appear pixelated.

It is suggested to center figures 4 and 6.

The labels on figure 7 are not legible. It is suggested to increase the font size or consider using an alternative visualization.

The names of the functional pathways in figure 8 are not legible.

Line 248: It is suggested to include the presence of species-specific bacterial pathogens in the analysis and discussion.

Author Response

Dear editor and reviewers

Thank you for dedicating your time to reviewing our manuscript (animals-2642524). We sincerely appreciate the valuable comments and suggestions provided by the reviewers, which have greatly contributed to the improvement of our work. We have diligently incorporated your kind and constructive feedback into the revised version of the manuscript, as evidenced by the itemized responses provided below and the corresponding revisions/corrections in the re-submitted files.

We are grateful for the opportunity to submit a revised copy of the manuscript and remain hopeful that it will be accepted for publication in animals.

Sincerely yours,

Guangwei Hu

On behalf of all co-authors

# Reviewer 2

Comment 1: It is recommended to include the following reference in the introduction. Lin, W., Ren, Z., Mu, C., Ye, Y., & Wang, C. (2020). Effects of elevated p CO2 on the survival and growth of Portunus trituberculatus. Frontiers in Physiology, 11, 750

Response 1: Thanks for your comment. The reference has been added in the revised version. Please see line 48-49.

Comment 2: Line 84: It is suggested to include abiotic data such as temperature, oxygen (saturation and concentration), and water turnover.

Response 2: the abiotic data, such as temperature, oxygen concentration and salinity were added. Please see the revised manuscript line 97-100.

Comment 3: Line 93: It is recommended to specify whether tissue and intestinal contents are included.

Response 3: For 16s rDNA sequencing the intestine tissue and contents are included. We have clarified this in the revised manuscript. Please see line 103-104.

Comment 4: Line 108: It is suggested to include details of the quality controls performed.

Response 4: The relevant information has been added. Please see our revised version (line125-131).

Comment 5: Line 111: It is recommended to provide the reference for the primers used.

Response 5: The reference for the primers used in this manuscript has been added. Please see line 122-124.

Comment 6: Line 140: It is suggested to include a table with the frequency of described findings. Please review possible artifacts due to sample processing or sectioning.

Response 6: Thank you for your suggestion. In this section, our objective is to assess the integrity of E. carinicauda's intestinal tissues under various treatments. The findings depicted in Figure 1 unequivocally demonstrate that the control group exhibited a normal and intact morphology of the intestinal tissue, whereas the AC74 and AC70 groups displayed deformation and exfoliation of epithelial cells from the basement membrane. It is worth noting that previous studies have commonly and acceptably presented the results of histological analysis using histological sections. Thanks again for this insightful comment.

References for review

[1] Cheng, C., Ma, H., Liu, G., et al. Toxic effects of cadmium exposure on intestinal histology, oxidative stress, microbial community, and transcriptome change in the mud crab (Scylla paramamosain). Chemosphere. 2023, 326, 138464

[2] Duan, Y., Wang, Y., Liu, Q., et al. Changes in the intestine barrier function of Litopenaeus vannamei in response to pH stress. Fish & Shellfish Immunology. 2019, 88, 142-149.

Comment 7: It is possible to include quantitative comparisons of 16S amplicon abundance by comparing the read counts of the analyzed samples in a supplementary figure.

Response 7: The 16S amplicon abundance of the analyzed samples has been added, please see line 176 and Table S1.

Comment 8: It is possible to include rarefaction curves of 16S amplicons for the different groups in a supplementary figure.

Response 8: Rarefaction curves has been added in the revised version. Please see line 181-183 and figure 2B.

Comment 9: The labels on figures 4, 5, and 6 appear pixelated.

Response 9: Thanks for your comment. Figures 4, 5, and 6 have been replaced. Please see the revised version.

Comment 10: It is suggested to center figures 4 and 6.

Response 10: Thanks for your suggestion. We adjusted the position of figure 4 and figure 6 according to your suggestion. Please see the revised version.

Comment 11: The labels on figure 7 are not legible. It is suggested to increase the font size or consider using an alternative visualization.

Response 11: Thanks for your suggestion, we replaced figure 7 in the revised version. Please see our revised manuscript.

Comment 12: The names of the functional pathways in figure 8 are not legible.

Response 12: Thanks for your comment, we replaced figure 8 in the revised version. Please see our revised manuscript.

Comment 13: Line 248: It is suggested to include the presence of species-specific bacterial pathogens in the analysis and discussion.

Response 13: We appreciate your valuable suggestion, as it significantly contributes to our comprehensive comprehension of the findings. The present study observed an augmentation in the abundance of Photobacterium subsequent to exposure to acidified seawater. Photobacterium, a marine bacterium belonging to the Vibrio family, encompasses certain members known for their pathogenic properties. However, no specific bacterial pathogens of Photobacterium targeting E. carinicauda have been reported thus far. Consequently, this study focuses on the discussion of abundance alterations at the genus level.

Round 2

Reviewer 1 Report

Thanks for your responses I am satisfied with your corrections.